# Effect of Azithromycin on Proinflammatory Cytokine Production in Gingival Fibroblasts and the Remodeling of Periodontal Tissue

**DOI:** 10.3390/jcm10010099

**Published:** 2020-12-30

**Authors:** Takatoshi Nagano, Takao Yamaguchi, Sohtaro Kajiyama, Takuma Suzuki, Yuji Matsushima, Akihiro Yashima, Satoshi Shirakawa, Kazuhiro Gomi

**Affiliations:** 1Department of Periodontology, Tsurumi University School of Dental Medicine, Yokohama 230-8501, Japan; kajiyama-sotaro@tsurumi-u.ac.jp (S.K.); suzuki-takuma@tsurumi-u.ac.jp (T.S.); matsushima-y@tsurumi-u.ac.jp (Y.M.); yashima-akihiro@tsurumi-u.ac.jp (A.Y.); shirakawa-satoshi@tsurumi-u.ac.jp (S.S.); gomi-k@tsurumi-u.ac.jp (K.G.); 2Aiko Yamaguchi Dental Clinic, Atsugi 243-0028, Japan; liberty@v006.vaio.ne.jp

**Keywords:** azithromycin, human gingival fibroblast, human periodontal ligament fibroblast, IL-6, IL-8, MMP-1, MMP-2

## Abstract

Previous reports have shown that azithromycin (AZM), a macrolide antibiotic, affects collagen synthesis and cytokine production in human gingival fibroblasts (hGFs). However, there are few reports on the effect of AZM on human periodontal ligament fibroblasts (hPLFs). In the present study, we comparatively examined the effects of AZM on hGFs and hPLFs. We monitored the reaction of AZM under lipopolysaccharide (LPS) stimulation or no stimulation in hGFs and hPLFs. Gene expression analyses of interleukin-6 (IL-6), interleukin-8 (IL-8), matrix metalloproteinase-1 (MMP-1), matrix metalloproteinase-2 (MMP-2), and Type 1 collagen were performed using reverse transcription-polymerase chain reaction (RT-PCR). Subsequently, we performed Western blotting for the analysis of the intracellular signal transduction pathway. In response to LPS stimulation, the gene expression levels of IL-6 and IL-8 in hGFs increased due to AZM in a concentration-dependent manner, and phosphorylation of nuclear factor kappa B (NF-κB) was also promoted. Additionally, AZM caused an increase in MMP-1 expression in hGFs, whereas it did not affect the expression of any of the analyzed genes in hPLFs. Our findings indicate that AZM does not affect hPLFs and acts specifically on hGFs. Thus, AZM may increase the expression of IL-6 and IL-8 under LPS stimulation to modify the inflammatory response and increase the expression of MMP-1 to promote connective tissue remodeling.

## 1. Introduction

We previously demonstrated the clinical utility of full-mouth as well as partial- mouth scaling and root planing (SRP) with the administration of the macrolide antibiotic azithromycin (AZM) [1,2,3,4]. Studies have indicated that while performing full-mouth SRP (FM-SRP) with AZM administration, inflammation was reduced and periodontal pockets improved rapidly within an extremely short period of time. This appears to be the result of the synergistic effect of the control of periodontopathic bacteria due to the antibacterial action of AZM and mechanical plaque control by SRP.

However, when other antibiotics were used in the same manner, results equivalent to or better than those of AZM could not be obtained [5], especially with regard to the reduction in inflammation of periodontal pockets during the early stage after treatment [6]. Thus, it appears that the effect of AZM is not limited to its antibacterial action, suggesting the possible involvement of another mechanism.

Some reports have indicated that AZM is effective in treating cyclosporine-induced gingival overgrowth [7,8,9]. In 2008, Kim et al. reported that AZM inhibits the proliferation of gingival fibroblasts and activations of matrix metalloproteinase-1 (MMP-1) and matrix metalloproteinase-2 (MMP-2). These metalloproteinases play a central role in collagen degradation. This may thereby reduce the volume of connective tissue constituents within the gingival tissue, resulting in amelioration of the gingival overgrowth associated with drug-induced fibrosis [10].

AZM and clarithromycin are known to regulate immune function and have been indicated to be particularly useful in treating pulmonary diseases, such as cystic fibrosis and chronic panacinar bronchitis [11,12]. A study using human bronchial epithelial cells reported that AZM’s efficacy against long-term chronic inflammation is due to increased interleukin-8 (IL-8) production [13]. Furthermore, it has been suggested that AZM increases IL-8 production in human gingival fibroblasts more than other macrolide antibiotics (i.e., erythromycin, josamycin), resulting in further promotion of neutrophil migration during the inflammatory response triggered by periodontal pathogens [14]. However, many points of these theories remain unclear.

The purpose of this study was to investigate the reaction of human gingival fibroblasts (hGFs) and human periodontal ligament fibroblasts (hPLFs) when FM-SRP is performed with AZM administration. Therefore, in the present study, we selected interleukin-6 (IL-6) and IL-8, cytokines involved in the regulation of the inflammatory response, to conduct experiments to examine the effects of AZM and cell behavior with hGFs and hPLFs stimulated by lipopolysaccharide (LPS). MMP-1, MMP-2, and Type I-Collagen were studied to investigate the effects of AZM on periodontal tissue remodeling. Further, we examined the effects of AZM on the intracellular signal transduction pathway of hGFs.

## 2. Materials and Methods

The present study was approved by the institutional review board of the School of Dental Medicine, Tsurumi University (approval number: 1035) and was conducted in accordance with their regulations.

HGFs and hPLFs were obtained from patients who were determined to require tooth extraction for orthodontic treatment or oral surgery upon visiting Tsurumi University Dental Hospital, and who also gave consent to participate in the experiment. HGFs and hPLFs were prepared as described previously [15,16].

Using cells in the third to sixth passages, we investigated the effects of AZM on hGFs and hPLFs.

The experiment protocol is demonstrated in Figure 1.

### 2.1. Cell Culture

Gingival tissue and periodontal ligament tissue was collected, finely chopped, and cultured, and the outgrowth cells of each were used as hGFs and hPLFs, respectively. Alpha-minimum essential media (α-MEM) (SIGMA) containing 10% fetal bovine serum (FBS) (ICN Biomedicals Inc., Solon, OH, USA) and 2% penicillin–streptomycin (Pn-St) (SIGMA, St. Louis, MO, USA) were used as serum and cultured at 37 °C in the presence of 5% CO_2_. A subculture was performed using 0.1% trypsin–0.04% EDTA (SIGMA).

### 2.2. MTS Assay (Cell Proliferation Assay)

To investigate the effects of AZM on the cell proliferation ability of hGFs and hPLFs, the cells were uniformly seeded on a 96-well plate with 1.0 × 10^4^ cells/well, and cultured in α -MEM containing 10% FBS and 1% Pn-St at 37 °C in the presence of 5% CO_2_.

At 1, 4, and 7 d after the addition of AZM (Pfizer Japan Inc., Tokyo, Japan), 20 μL of Cell Titer 96 Aqueous One Solution Reagent (Promega, Madison, WI, USA) was added and incubated for 60 min. Absorbance was subsequently measured at a wavelength of 490 nm.

### 2.3. Reverse Transcription-Polymerase Chain Reaction (RT-PCR)

Both hGFs and hPLFs were seeded separately in 6-well plates with a cell count of 1.0 × 10^4^ cells/well, and the culture medium was replaced every two days.

After confluence was observed, the experiment was conducted by transferring the cells to a stimulating medium, to which a sample was added 24 h after exchanging with α-MEM without the addition of serum. The stimulation medium was created by adding an LPS (LPS; *Escherichia coli* 0111: B4, SIGMA) and AZM to α-MEM to a final concentration of 0 or 5 μg/mL and 0 to 100 μg/mL, respectively, at 37 °C after pre-incubation in a 5% CO_2_ environment. The gene expression levels of the inflammatory cytokines IL-6 and IL-8 were examined after a specific period. The gene expression levels of MMP-1, MMP-2, and Type I-Collagen, which are involved in connective tissue remodeling, were examined 1, 4, and 7 days after AZM was added without LPS.

Total RNA was extracted from cultured cells using RNAqueous (Applied Biosystems, Austin, TX, USA), and cDNA was prepared using Ready-to-go You-Prime First-Strand Beads (GE Healthcare Life Sciences, Piscataway, NJ, USA). Glyceraldehyde 3-phosphate dehydrogenase (GAPDH) (Clontec, Palo Alto, CA, USA) was used as the internal standard.

RT-PCRs were performed using specific primers and Taq DNA polymerase (Takara Bio Inc., Shiga, Japan) (Applied Biosystems/Gene Amp PCR System 9700). PCR products were detected by performing 4.5% polyacrylamide gel electrophoresis (PAGE), staining with ethidium bromide, and subsequent exposure to ultraviolet light. The results were recorded.

The gene sequences of each primer are shown in Figure 2.

### 2.4. Western Blotting

Total protein was extracted from the cultured hGFs under the same conditions maintained for total RNA extraction, and an extraction buffer containing 2% sodium dodecyl sulfate (SDS) and 0.1 M dithiothreitol (DTT) was used. After pretreatment, 10% SDS-PAGE was performed to separate the proteins and transfer them to the polyvinylidene difluoride (PVDF) membrane (Bi-Rad Laboratories, Hercules, CA, USA). After blocking with bovine serum albumin (BSA), an antibody specific for each protein (nuclear factor kappa B (NF-κB), phospho NF-κB, p38 mitogen-activated protein kinase (MAPK), phospho p38 MAPK, Jun N-terminal kinase (JNK), and phospho JNK; Cell Signaling Technology, Danvers, MA, USA) was used as the primary antibody, and horseradish peroxidase (HRP)-labeled anti-rabbit immunoglobulin G (IgG) antibody (Cell Signaling Technology) was used as the secondary antibody. Immunological detection was performed using enhanced chemiluminescence (ECL) Western blotting detection reagents (GE Healthcare Life Sciences Corp, Verdesian, MA, USA), and the activation of intracellular signal transduction molecules associated with the expression of inflammatory cytokines was investigated. β-actin (Cell Signaling Technology) was used as the internal standard.

### 2.5. Data Analysis and Statistical Processing

Based on the results of our pilot study, we calculated the number of samples in this study using the formula *n* = 16(SD/Δ)^2^, where SD is the average standard deviation between the two groups and Δ is the difference between the average values of the two groups. It was calculated at a significance level α of 0.05 and power of 80% (β = 0.2). As a result, all experiments were performed with *n* = 3. All the experiments were performed at least three times.

The results of RT-PCR electrophoretic imaging were standardized using GAPDH with a Densitograph Lane & Spot Analyzer (ATTO, Tokyo, Japan), and semi-quantification was performed. Students t-tests were used for data analysis of the MTS assay, and statistical significance was defined as *p* < 0.05.

## 3. Results

### 3.1. MTS Assay (Cell Proliferation Assay)

MTS assay results showed that AZM did not statistically affect the proliferative activity of hGFs and hPLFs (Figure 3 and Figure 4).

### 3.2. IL-6 and IL-8 Gene Expression

In hGFs, the gene expression levels of IL-6 and IL-8 were found to increase due to stimulation with LPS (Figure 5), whereas no changes in IL-6 and IL-8 gene expression levels were observed with the addition of AZM alone (Figure 6). Thus, it was demonstrated that AZM by itself does not amplify the gene expression levels of IL-6 and IL-8.

Moreover, in the case of hGFs, the gene expression levels of IL-6 and IL-8 increased in a concentration-dependent manner due to AZM in the presence of LPS (Figure 7). However, hPLFs did not show any changes with regard to IL-6 and IL-8 gene expression (Figure 8).

### 3.3. Western Blotting

Western blot of hGFs showed an increase in the phosphorylation of NF-κB. However, no change was observed in the phosphorylation patterns of p38 MAPK and JNK in the presence of both LPS and AZM (Figure 9).

### 3.4. Gene Expression Levels of MMP-1, MMP-2, and Type I-Collagen

Increased MMP-1 gene expression levels were observed in hGFs due to the effects of AZM (Figure 10), whereas no significant changes were observed in the gene expression levels of MMP-2 and Type I-Collagen. In hPLFs, no prominent change was observed in the gene expression levels of MMP-1, MMP-2, and Type I-Collagen (Figure 11).

## 4. Discussion

AZM is a macrolide antibiotic with a long half-life and a wide antibacterial spectrum. It has been demonstrated to be useful in periodontal treatment due to its ability to inhibit biofilm formation [17]. AZM is characterized by phagocyte delivery, which allows for selective uptake by phagocytic cells, causing accumulation at the site of inflammation. AZM accumulated locally in the inflamed gingival tissue temporarily increased the expression of IL-6 and IL-8, which further promoted neutrophil migration. It is possible that there may be a mechanism by which high concentrations of AZM are selectively accumulated in the periodontal tissue at an early stage as a result of the aforementioned synergistic effects.

When performing periodontal treatment under AZM administration, the antibacterial action of AZM may facilitate the control of periodontopathic bacteria by creating a microenvironment in which a high concentration of AZM selectively gathers locally in the periodontal tissue at an early stage. As a result, the inhibitory or destructive action of AZM against biofilm formation may ameliorate microbial accumulation in the periodontal pocket. It has also been reported that AZM does not significantly affect cell adhesion ability and human epithelial cell death [18]. The results of the present study indicate that AZM does not affect the cell proliferation activity of either hGFs or hPLFs, confirming that AZM can be safely administered in periodontal treatment.

In the present study, the gene expression of IL-6 and IL-8 of hGFs increased in a concentration-dependent manner due to the combined effects of AZM and LPS. Furthermore, the use of AZM alone did not affect the gene expression of IL-6 and IL-8. Moreover, hPLFs showed no significant changes in IL-6 and IL-8 gene expression. The ability of AZM to modify the inflammatory response is a characteristic phenomenon in hGFs, and the use of AZM alone does not appear to be able to modify or regulate the response. Thus, these results suggest that the action of AZM at the site of inflammation might temporarily increase the inflammatory response in the presence of LPS. However, in a study using KB cells, a human oral epithelial cell line, it was reported that AZM inhibited the expression of IL-8 induced by LPS [19]. Based on these findings, the effects of AZM could differ, depending on the target cell.

In 2015, Doyle et al. reported that the expression of the inflammatory cytokines IL-6, IL-8, monocyte chemoattractant protein-1 (MCP-1), and growth-regulated oncogene (GRO), previously induced by LPS derived from *Porphyromonas gingivalis* against hGFs, was reduced due to the effects of AZM [20]. In that study, 10 μg/mL of AZM was used, and the results were recorded 24 h after the addition of AZM. In the present study, however, the maximum concentration of AZM used was 100 μg/mL, the stimulation time was particularly short (2 h), and the experimental conditions differed. Based on these findings, it appears that AZM may play a role in strengthening the inflammatory response during the very early stage in the presence of LPS, and subsequently reducing the inflammatory response in latter stages. Thus, its effects may be present for a long period. However, since hGFs may behave and react differently depending on the intracellular concentration of AZM and the environment of action, further research in this area is needed.

Moreover, in the present study, phosphorylation of NF-κB was also promoted when LPS and AZM acted simultaneously, compared with when stimulation with LPS alone was performed, whereas there were no significant changes in the phosphorylation of p38 MAPK and JNK. These findings indicate that an increased production of inflammatory cytokines in hGFs may be caused by the effects of AZM due to LPS stimulation. This may be mediated by the activation of the signal transduction pathway of NF-κB. A previous report showed that the simultaneous action of AZM and tumor necrosis factor-alpha (TNF-α) in infant tracheal aspirate cells inhibited NF-κB signal activation [21]. It has also been reported that when the effects of AZM were investigated using murine macrophage cell lines, the NF-κB signal was inhibited [22]. A study using human peripheral blood mononuclear cells reported that AZM had a major effect on JNK expression [23]. Furthermore, in 2015 it was reported that AZM inhibited the phosphorylation of p38 MAPK and extracellular-signal-regulated kinase (ERK) signaling pathways in human gingival epithelial cells activated by TNF-α stimulation [24]. The reason for non-concurrence of the results of the present study with those of the previous one may be because the effects of AZM on the intracellular signal transduction pathway may differ, depending on factors such as the degree and timing of inflammation, tissue and cell type, patient age, and study designs. Extracellular vesicles are currently known to be the main source of proinflammatory cytokines. Especially, microvesicles may function as strong regulators of the immune system [25]. Given that the microvesicle function and network of signal transduction pathways in cells are complicated, further studies are necessary to clarify these findings.

In 1995, it was reported that AZM improved cyclosporine-induced gingival overgrowth [26], and continued research indicated that said effect was due to the promotion of phagocytosis of fibroblasts by AZM [27]. The results of the present study indicate that AZM acted specifically on hGFs and increased the expression of MMP-1. This suggests that the action of MMP-1 produced by gingival fibroblasts on collagen promoted its decomposition, thereby inducing periodontal tissue remodeling. This mechanism might explain the clinical findings indicating that gingival retraction results in the early and dramatic reduction of periodontal pockets when performing FM-SRP during AZM administration. However, since a significant reaction to the expression of MMP-1 was not observed in hPLFs, it appears that this unique effect of AZM was specific to the gingival tissue.

Some researchers are of the view that the clinical results of FM-SRP without AZM administration are comparable to those in the case of conventional SRP [28]. However, it has been previously reported that the formation of human osteoclasts is inhibited due to the effects of AZM [29]. Furthermore, performing FM-SRP with AZM administration can inhibit the production of inflammatory mediators and increase total body temperature [30,31]. Hence, applying AZM to periodontal treatment may produce various secondary benefits and demonstrate efficacy in the clinical setting.

These findings suggest that when FM-SRP is performed with AZM administration, AZM amplifies the gene expression of IL-6 and IL-8, which are involved in neutrophil migration in the inflamed gingival region. This contributes to the amelioration of inflammation during the early stage. This suggests the existence of a mechanism by which the periodontal pockets are reduced as a result of the remodeling of the gingival connective tissue due to the increased gene expression of MMP-1.

## 5. Conclusions

The results of the present study suggest that in the presence of LPS, AZM may activate the NF-κB signaling dental pathway in hGFs, resulting in the increased production of inflammatory cytokines. AZM may also increase the expression of MMP-1 in hGFs, thereby promoting gingival connective tissue remodeling.

## Figures and Tables

**Figure 1 jcm-10-00099-f001:**
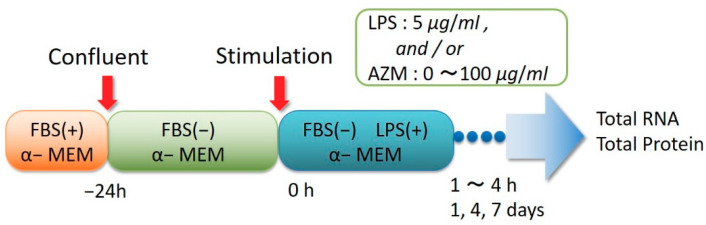
Experimental protocol.

**Figure 2 jcm-10-00099-f002:**
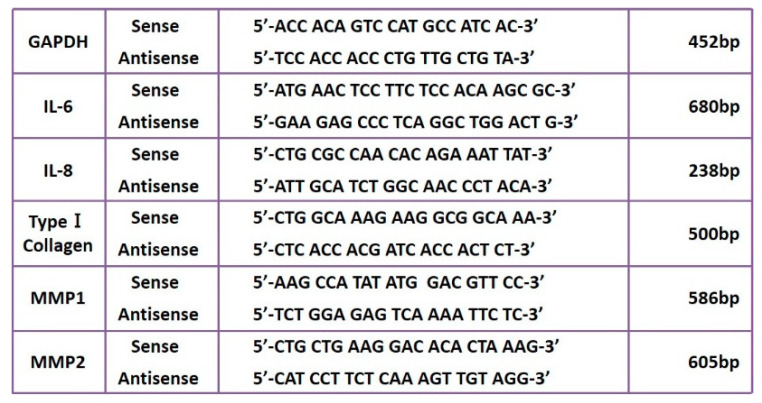
PCR primers.

**Figure 3 jcm-10-00099-f003:**
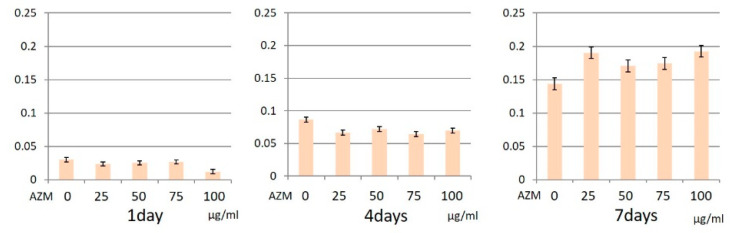
Cell proliferation ability of human gingival fibroblasts (hGFs) with azithromycin (AZM). These experiments were performed in triplicate, and data are presented as the mean ± SD.

**Figure 4 jcm-10-00099-f004:**
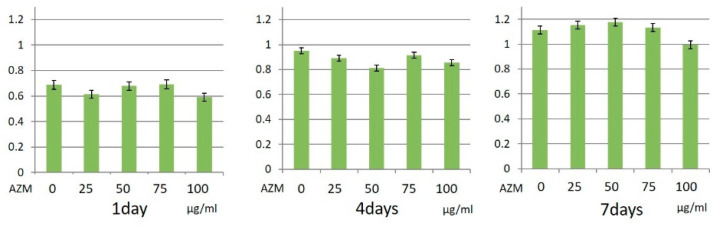
Cell proliferation ability of human periodontal ligament fibroblasts (hPLFs) with AZM. These experiments were performed in triplicate, and data are presented as the mean ± SD.

**Figure 5 jcm-10-00099-f005:**
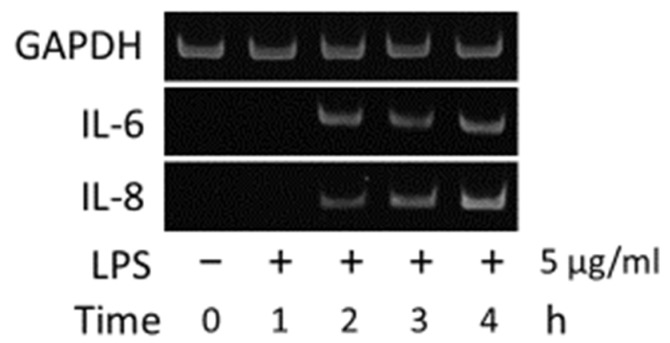
LPS enhances IL-6 and IL-8 gene expression in hGFs.

**Figure 6 jcm-10-00099-f006:**
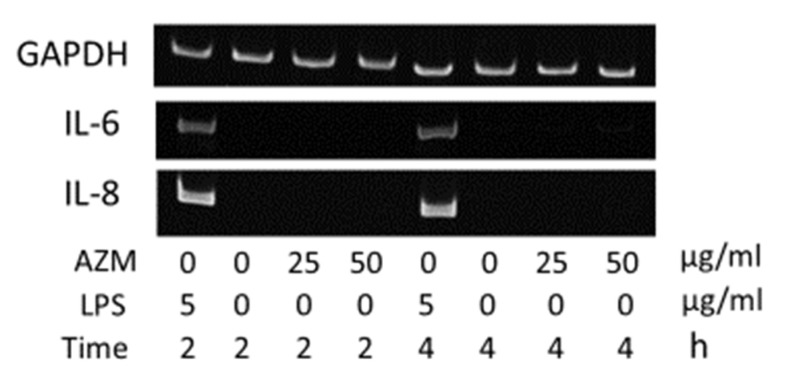
AZM alone does not enhance gene expression of IL-6 and IL-8 in hGFs.

**Figure 7 jcm-10-00099-f007:**
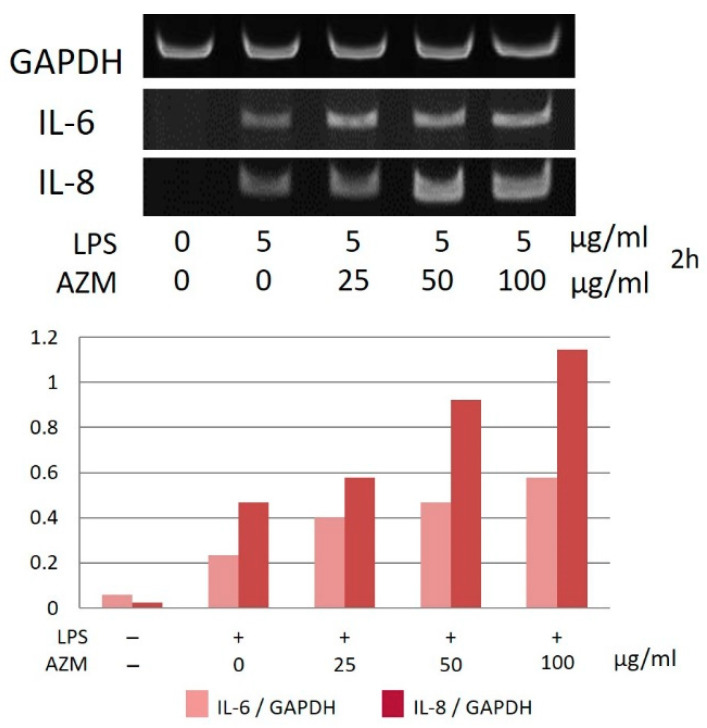
Gene expression of IL-6 and IL-8 in hGFs in the presence of LPS and AZM. The most typical data were shown.

**Figure 8 jcm-10-00099-f008:**
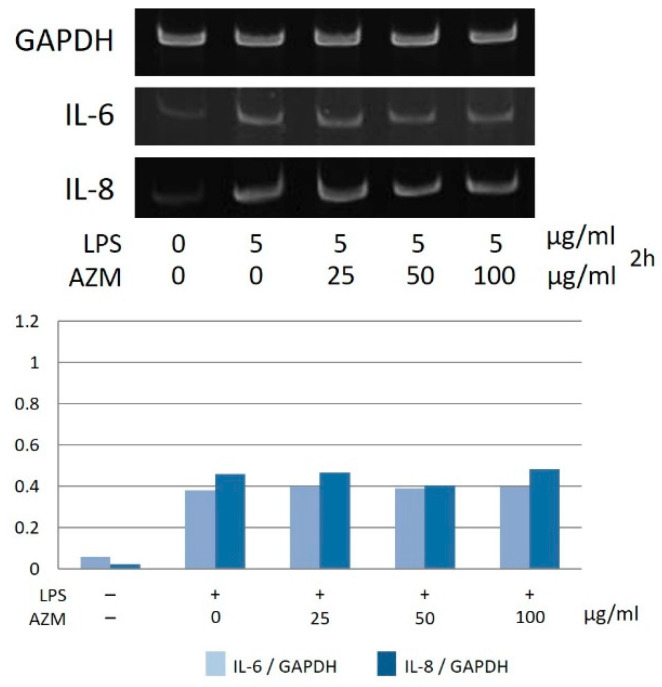
Gene expression of IL-6 and IL-8 in hPLFs in the presence of LPS and AZM. The most typical data were shown.

**Figure 9 jcm-10-00099-f009:**
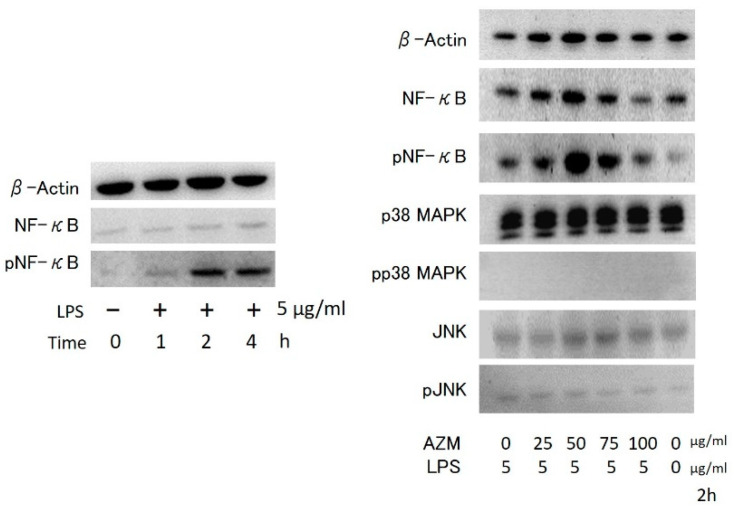
Effects on intracellular signaling molecules in hGFs in the presence of LPS and AZM.

**Figure 10 jcm-10-00099-f010:**
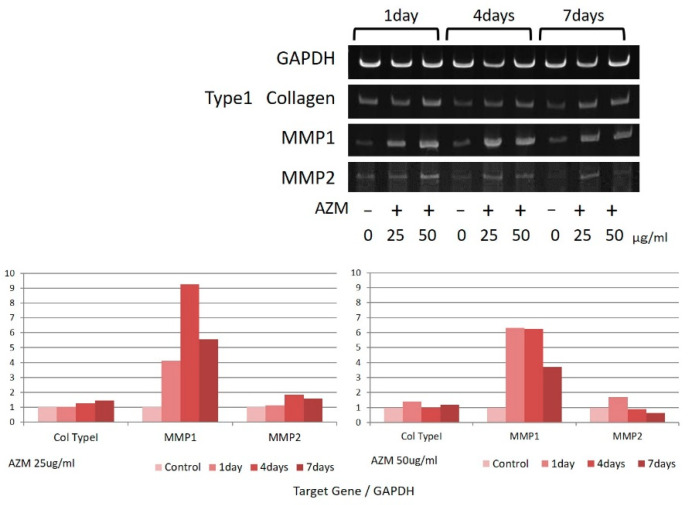
Gene expression of Type I-Collagen, MMP-1, and MMP-2 in hGFs due to the influence of AZM. The most representative data were shown.

**Figure 11 jcm-10-00099-f011:**
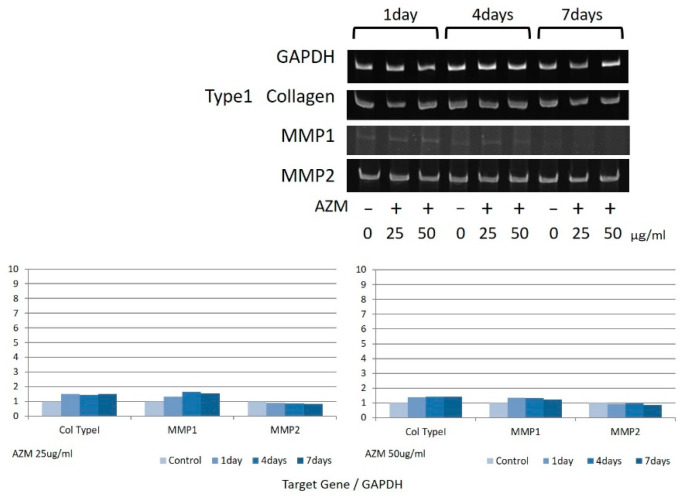
Gene expression of Type I-Collagen, MMP-1, and MMP-2 in hPLFs due to the influence of AZM. The most representative data were shown.

## Data Availability

All relevant data are available upon request. Please address requests to Takatoshi Nagano.

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
