# Peer review of "Effect of Azithromycin on Proinflammatory Cytokine Production in Gingival Fibroblasts and the Remodeling of Periodontal Tissue"

_jcm, 2020, doi:10.3390/jcm10010099_

Round 1
Reviewer 1 Report
Overall really good study on a a novel and pertinent topic.
My concern is with the material and method section: please include the sample size, sample size calculation, inclusion, exclusion criteria. Who conducted the study, how many examiners, were they calibrated, for how long the study lasted etc.
Author Response
Dear Reviewer,
We would like to thank you for your prompt and valuable comments on the revised version of our manuscript. We have consulted with a statistician and have addressed all comments, as indicated in the attached pages, and we hope that the explanations are satisfactory.
We hope that the revised version of our manuscript is now suitable for publication in Journal of Clinical Medicine: A special issue of Prevention and Treatment of Periodontitis.
Sincerely Yours,
Takatoshi Nagano
Q : Sample size
→ We described that how to determine about sample size in material and method section. In addition, we have already explained that who conducted the study, how many examiners, in author contributions section. We performed this experiment from 2011 to 2017.
[Editor1]Remark: Thank you for availing our Manuscript Insurance service. We have checked the responses to the comments by the reviewers and have made minor revisions. Feel free to contact us in case of any further queries. We will be happy to help you in the best way possible.

Reviewer 2 Report
In this article, the authors attempted to assess the effect of azithromycin on the production of selected pro-inflammatory cytokines. The article is written very well. Regardless of this, I have a few comments that the authors should take into account:
1. introduction is properly prepared and the purpose of the work is clearly defined.
2. Methods - please add more details regarding the collection of cells from patients.
3. The figures are of poor quality.
4. Table 1 is illegible.
5. Discussion - extracelular vesicles are currently known to be the main source of proinflammatory cytokines (https://doi.org/10.3389/fimmu.2018.02723). This aspect should be discussed e.g. in the context of further research. In discussion the authors may use the above-cited paper.
Author Response
Dear Reviewer,
We would like to thank you for your prompt and valuable comments on the revised version of our manuscript. We have consulted with a statistician and have addressed all comments, as indicated in the attached pages, and we hope that the explanations are satisfactory.
We hope that the revised version of our manuscript is now suitable for publication in Journal of Clinical Medicine: A special issue of Prevention and Treatment of Periodontitis.
Sincerely Yours,
Takatoshi Nagano
Major corrections are as follows:
- We have added the aim of this study in introduction section.
- We have added the details of cells collection procedure in material and methods section.
- We have been corrected the figures and figure legends.
- Table 1 have been corrected, according to suggestion.
- We have added your recommendation paper in discussion section.
- Typographical errors have been corrected.
[Editor1]Remark: Thank you for availing our Manuscript Insurance service. We have checked the responses to the comments by the reviewers and have made minor revisions. Feel free to contact us in case of any further queries. We will be happy to help you in the best way possible.

Round 2
Reviewer 2 Report
The authors have addressed all the comments of the reviewer and revised the manuscript accordingly.
Congratulations and happy New Year.